# High-Temperature Polylactic Acid Proves Reliable and Safe for Manufacturing 3D-Printed Patient-Specific Instruments in Pediatric Orthopedics—Results from over 80 Personalized Devices Employed in 47 Surgeries

**DOI:** 10.3390/polym16091216

**Published:** 2024-04-26

**Authors:** Grazia Chiara Menozzi, Alessandro Depaoli, Marco Ramella, Giulia Alessandri, Leonardo Frizziero, Adriano De Rosa, Francesco Soncini, Valeria Sassoli, Gino Rocca, Giovanni Trisolino

**Affiliations:** 1Unit of Pediatric Orthopedics and Traumatology, IRCCS Istituto Ortopedico Rizzoli, 40136 Bologna, Italy; graziachiara.menozzi@ior.it (G.C.M.); marco.ramella@ior.it (M.R.); gino.rocca@ior.it (G.R.); 2Rizzoli Sicilia Department, IRCCS Istituto Ortopedico Rizzoli, 90011 Bagheria, Italy; alessandro.depaoli@ior.it; 3Department of Industrial Engineering, Alma Mater Studiorum University of Bologna, 40136 Bologna, Italy; giulia.alessandri5@unibo.it (G.A.); leonardo.frizziero@unibo.it (L.F.); adriano.derosa3@unibo.it (A.D.R.); 4Unit of Hygiene, Epidemiology and Emergency Management, IRCCS Istituto Ortopedico Rizzoli, 40136 Bologna, Italy; francesco.soncini@ior.it; 5Pharmacy Service, IRCCS Istituto Ortopedico Rizzoli, 40136 Bologna, Italy; valeria.sassoli@ior.it

**Keywords:** polylactic acid, high-temperature polylactic acid, annealing, sterilization, in vivo, orthopedic surgery, pediatrics, 3D-printing, patient-specific instrument, Fused Deposition Modeling

## Abstract

(1) Background: Orthopedic surgery has been transformed by 3D-printed personalized instruments (3DP-PSIs), which enhance precision and reduce complications. Hospitals are adopting in-house 3D printing facilities, using cost-effective methods like Fused Deposition Modeling (FDM) with materials like Polylactic acid (PLA) to create 3DP-PSI. PLA’s temperature limitations can be overcome by annealing High-Temperature PLA (ann-HTPLA), enabling steam sterilization without compromising properties. Our study examines the in vivo efficacy of ann-HTPLA 3DP-PSI in pediatric orthopedic surgery. (2) Methods: we investigated safety and efficacy using ann-HTPLA 3DP-PSI produced at an “in-office” 3D-printing Point-of-Care (3DP-PoC) aimed at correcting limb deformities in pediatric patients. Data on 3DP-PSI dimensions and printing parameters were collected, along with usability and complications. (3) Results: Eighty-three ann-HTPLA 3DP-PSIs were utilized in 33 patients (47 bone segments). The smallest guide used measured 3.8 cm^3^, and the largest measured 58.8 cm^3^. Seventy-nine PSIs (95.2%; 95% C.I.: 88.1–98.7%) demonstrated effective use without issues. Out of 47 procedures, 11 had complications, including 2 infections (4.3%; 95% CI: 0.5–14.5%). Intraoperative use of 3DP-PSIs did not significantly increase infection rates or other complications. (4) Conclusions: ann-HTPLA has proven satisfactory usability and safety as a suitable material for producing 3DP-PSI in an “in-office” 3DP-PoC.

## 1. Introduction

Three-dimensional printing has emerged as a transformative tool in orthopedics, heralded for its ability to streamline surgical procedures. The widespread adoption of 3D printing from anatomical models to implants to templates and cutting guides for fractures or osteotomies is well-documented, with converging evidence of significant benefits [1]. Reduced operating time, fluoroscopy, bleeding, and complications while improving surgical precision, marking a substantial advance in orthopedic practice [2,3,4,5], are among the most known and documented benefits. Of notable significance are the advantages provided by 3D-printed patient-specific instruments (3DP-PSI) in pediatric orthopedic surgery [6,7]. These innovative tools facilitate complex procedures such as bone cuts in multiplanar osteotomies and bone tumor surgeries, marking a significant advancement in orthopedic care.

Despite its potential, the widespread adoption of 3D printing in orthopedics has been impeded primarily by cost considerations. Overcoming this hurdle necessitates a paradigm shift toward internalizing the production process for 3DP-PSI within healthcare facilities and using low-cost 3D-printing technology [7]. Hence, more hospitals are adopting “in-hospital” 3D-Labs and “in-office” 3D-Printing point-of-care (3DP-PoC) setups to enhance accessibility and user-friendliness, notably cutting management costs. The predominant printing methods employed in these settings include Fused Deposition Modeling (FDM), Selective Laser Sintering (SLS), and Stereolithography (SLA) [8].

Fused Deposition Modeling (FDM) 3D printers offer a versatile and cost-effective solution for manufacturing objects “in-office” with minimal resources. However, cost-effectiveness should never compromise patient safety or adherence to healthcare standards, including compatibility with sterilization processes. Ensuring feasibility requires verifying that 3DP-PSI produced in healthcare settings can undergo sterilization without compromising geometric properties [9,10]. Although FDM has been extensively used for 3D-printed models and tools in orthopedic surgery, challenges remain in surface quality, layer fusion, and print defects impacting functionality.

Optimizing the print quality of FDM-manufactured objects to meet specific (bio)mechanical requirements naturally involves selecting the most suitable material. While materials like metal offer excellent quality and properties, they are costly, especially for one-time personalized print, and incompatible with entry-level FDM printers [11]. Optimizing processing parameters is another option. This includes adjusting parameters like strand diameter, layer height, infill density, printing and building platform temperatures, and cooling rate [12,13].

Modifying the shape is an alternative, but it risks increasing the bulk of the cutting guide, potentially causing issues such as impingement with soft tissues and compromising adhesion between the cutting mask and the skeletal substrate.

Among the plethora of available materials, polylactic acid (PLA) is a promising candidate. Its affordability, ready availability, lack of toxic emissions during printing, ease of printing, and renewable sourcing render it ideal for 3D printing of sterilizable 3DP-PSI in an “in-office” setting.

However, PLA’s susceptibility to temperature, with a glass transition threshold of 60 °C, poses a potential challenge for steam sterilization, the predominant technique employed in hospitals.

To enhance the temperature resistance of PLA in 3D-printed tools, numerous companies have introduced modified variants of this material. Among them, high-temperature PLA (HTPLA) stands out as a promising option for creating 3D-printed objects that can endure steam sterilization without substantial changes to the shape, mechanical strength, biocompatibility, and other chemical–physical properties [14,15].

Companies developing high-temperature 3D printer filaments have not disclosed their exact methods, but many recommend annealing the printed objects. Annealing, a technique borrowed from metalworking, involves heating the part to a precise temperature to induce recrystallization without melting. This rearrangement at the molecular level is crucial for enhancing temperature resistance and mechanical strength. HTPLA, a PLA filament with a crystallization compound, stabilizes at higher temperatures via annealing, a thermal process that realigns constituent chains, resulting in a more ordered molecular structure akin to crystalline polymers. Controlled heating, targeting the glass transition temperature (Tg), optimizes this alignment process. Unlike crystalline materials, amorphous polymers experience a nuanced softening near Tg, marking the transition from a solid to a partially liquid state [16].

After annealing, HTPLA increases its heat resistance so it can be autoclaved. To provide context, other commonly used polymers for 3D printing, such as acrylonitrile butadiene styrene (ABS), have similar properties to PLA, are relatively flexible and tough and resistant to water degradation and temperatures ranging from −20 °C to 80 °C. However, they are not biodegradable; they can release harmful vapors during the printing process and cannot be annealed to improve their properties [17].

Despite several in vitro studies demonstrating the safety and effectiveness of annealed HTPLA (ann-HTPLA), there are few in vivo studies proving its utilization [18].

This study aims to document our preliminary experience in using ann-HTPLA 3D-printed patient-specific instruments (3DP-PSI), customized for pediatric patients, produced at an “in-office” 3DP-PoC.

## 2. Materials and Methods

### 2.1. Study Design

This interim analysis examines the safety and effectiveness of 3DP-PSI made from ann-HTPLA using FDM-3D printing within an “in-office” 3DP-PoC setting. It is part of a broader ongoing clinical trial (NCT05700526) dedicated to thoroughly investigating the management and production challenges associated with surgical intervention aids, including sterilizable 3DP-PSI, for pediatric patients with bone disorders and deformities requiring complex osteotomies and customized bone allografts. Specifically, interventions include the utilization of 3DP-PSI and the creation of personalized bone allografts facilitated by the patient–donor matching via Virtual Surgical Planning (VSP) supported by precise bone graft preparation aided by sterilizable graft-specific tools fabricated via 3D printing. Additional information on study design, eligibility criteria, recruitment procedures, and patient characteristics has been previously documented and is available in this study’s protocol [19,20].

### 2.2. Design Method

The methods for generating the virtual skeletal patient-specific 3D model from CT (Computer Tomography) scan via the segmentation process and simulating corrective osteotomies have been previously detailed by the authors in multiple articles [21,22]. Once we have the skeletal segment model of the patient and have simulated the desired correction, identifying Kirschner wires (K-wires) and slot positioning, we design the PSIs (generally templates and cutting guides) to meet the surgeon’s specifications and the anatomical constraints of the surgical site using Blender (Blender Foundation, Amsterdam, the Netherlands) for the surgical planning phase and Creo Parametric (PTC, Boston, MA, USA) for the cutting guide design phase (Figure 1).

We have empirically shaped the template and established certain dimensions for the design of the masks that have proven effective after heat treatment. For example, we secure 3DP-PSIs to bones using stainless steel K-wires, preferably 2.5 mm in diameter, for long bones such as the femur, tibia, humerus, and forearm bones. This necessitates creating 3 mm diameter holes to accommodate the 2.5 mm wires, ensuring smooth insertion without compromising the precision of the directional guidance provided by the guide. For cutting guides, we ensure the slot accommodates variable-width oscillating saw blades (Performance Series Blades—Stryker Co., Portage, MI, USA), ranging from 10 mm wide and from 0.38 mm to 0.77 mm thick. This design guarantees proper blade movement without excessive vibration or mask damage while maintaining the correct blade direction. Concerning the thickness of the cutting guide, our observation indicates that an offset of approximately 3 mm from the bone surface is optimal. This offset minimizes interference with the soft tissues in terms of bulk while maintaining effectiveness (Figure 2).

### 2.3. Fabrication Methods

All the guides were designed and exported to Standard Triangulation Language (.stl) file format using the CAD (Computer Aided Design) software Creo Parametric 10.0, used to design the guides as said before and printed with the Qidi i-Mate S FDM 3D-printer (Zhejiang Qidi Technology Co., Wenzhou, Zhejiang, China), with printing plane of the dimensions of 270 mm × 200 mm × 200 mm, a standard 0.4 mm nozzle, and 0.2 mm layer height, with a speed of about 60 mm/s. The prints were made in HTPLA (PLA Crystal Clear, Fillamentum Manufacturing Czech s.r.o., Hulin, Czech Republic), with an extrusion temperature of 220 °C and with the print bed heated to 60 °C (Table 1).

All parts were printed with 100% infill to avoid air voids, obtain the best possible mechanical properties, and help make the printed parts as less anisotropic as possible.

Moreover, to ensure proper adhesion of the initial layer, a reduced printing speed was employed. Additionally, a 20% overlap is configured to enhance the connection between perimeters and infill (for the printing parameters, please refer to Table 2).

For each printed guide, supports were positioned during print preparation when needed, and the best print position was evaluated to avoid affecting the surface of the bone-support mask and the parts with holes and saw slots (Figure 3).

Once printed, the masks were heat-treated. Annealing cycle characteristics (Table 3) were studied in previous studies [20,23,24]. After performing the thermal treatment, any remaining supports are removed, and the guide is checked for wire and saw blade passage before being sent for sterilization (see Appendix A).

### 2.4. Sterilization Process

The hospital sterilization center complies with UNI EN ISO 17665-1 [25] and ISO/TS 17665-2 standards [26] (see Appendix A). In particular, the sterilization process for 3DP-PSIs includes manual washing, control, and packaging in a controlled contamination environment, steam sterilization under pressure (134 °C temperature, 3 bar pressure, 7 min exposure time), cooling in a controlled contamination environment, followed by physical cycle checks and assessment of packaging integrity. The device is then registered in the medical device register, and all necessary documentation for device traceability is printed. The device is now ready for use in surgery (Figure 4).

### 2.5. Data Analysis

Patients were assigned numerical codes, and their data were input into Excel (Microsoft, Redmond, WA, USA) and SPSS (version 22.0; SPSS, Chicago, IL, USA). Continuous data were presented as mean ± standard deviation (SD) and range, while categorical and ordinal data were presented as raw numbers and proportions with a 95% confidence interval (C.I.). Normality was assessed using the χ2 test for categorical variables and the Kolmogorov–Smirnov test for continuous variables.

In determining the dimensions of the cutting guides, we accounted for the diverse applications and intricate organic shapes inherent in anatomical forms. To establish a reference system, we aligned with the subdivision of the three cardinal anatomical planes recognized in medical terminology. Height was defined as the dimension parallel to the diaphysis and situated within the coronal plane; width was designated as the dimension perpendicular to the diaphysis and positioned within the sagittal plane; depth was identified as the dimension perpendicular to the diaphysis and situated within the transverse plane. An illustrative example of these dimensions is provided in Figure 5.

The volume of the cutting guide was also taken into account.

The assessment included an evaluation of printing times, filament usage, and the number of printing layers for each mask. It is important to note that while geometric dimensions are specified, the actual printed volume differs due to the additional material required to support the printed part during fabrication.

We assessed the guide’s usability by examining its performance during surgery, identifying any manufacturing defects or mechanical damage, and gathering feedback from the surgeon. Peri-operative complications were assessed using the modified Clavien Dindo Sink classification by Dodwell et al. (mCDS) and compared to a cohort of children undergoing complex surgical interventions aided by VSP only, without utilizing 3D-PSIs [27]. Group differences were analyzed using Fisher’s exact test for categorical variables, considering a *p*-value < 0.05 as statistically significant.

## 3. Results

Between March 2018 and March 2023, 110 surgeries were planned and conducted using Virtual Surgical Planning (VSP) for 91 children. The 3D-printed Patient-Specific Instruments (3DP-PSIs) were utilized in 36 cases, with 33 patients undergoing scheduled surgery. These procedures addressed a total of 47 bone segments, including nine cases where patients needed bilateral and/or multilevel osteotomies. In the remaining three cases, the planned surgery and use of the manufactured PSIs were not feasible due to the following reasons: (1) A pre-operative fracture at the site of operation rendered the designed cutting guide unusable; (2) Multiple postponements due to the patient’s illness caused the patient to outgrow the originally designed cutting guide; (3) The third patient was unreachable on the scheduled operation day and dropped out of surgery and this study’s protocol.

Depending on the type and location of the surgical procedure, the usage of 3DP-PSIs varied from one to four per operation, leading to a total of 83 ann-HTPLA 3DP-PSIs produced, processed, sterilized, and utilized in the operating room.

The fabricated 3DP-PSI have been utilized in surgical interventions to address deformities in both upper and lower limbs, encompassing a diverse range of bone segments and surgical sites. The mean geometric dimensions and printing parameters are reported in Table 4.

In our analysis, we encountered 3DP-PSIs of diverse sizes, each exhibiting significantly different printing times. We were dealing mainly with deformities of the long bones of the lower limb, which proved, as one might have suspected, to need more material and 3D printing time, as well as to be larger in size. Currently, the smallest guide employed measured 3.8 cm^3^ in volume, 29.2 mm in height, 32.5 mm in length, and 15.9 mm in depth and was used to perform a corrective valgus osteotomy of the right distal humerus in a six-year-old child, while the largest guide registered 58.8 cm^3^ in volume, 64.2 mm in height, 54.4 mm in length, and 28.0 mm in depth was used for a corrective varus osteotomy of proximal femur in a 13-year-old girl.

Seventy-nine PSIs (95.2%; 95% C.I.: 88.1–98.7%) were used effectively with ease, demonstrating good efficacy and overall satisfaction from the surgeon’s perspective, without any reported issues regarding the positioning and performance of the instrument by the surgeons. Only four guides encountered problems. In the first case, the cutting guide suffered significant deformation, rendering it unusable due to improper positioning on the bone. This deformation likely stemmed from either the patient’s growth between the planning phase and the surgery or the guide’s exposure to double sterilization following a prior surgical postponement. In the second and third cases, the mask was fractured while being positioned on the bone (Figure 6). In both cases, the breakage occurred parallel to the 3D printing plane. In one instance, “delamination” of adjacent layers was evident, likely worsened by the high humidity prevailing during the summer in which surgery was performed, despite all required drying precautions being taken. Conversely, in the other case, a clean break occurred, likely as a result of excessive pressure during positioning. Lastly, in the fourth case, the mask was deemed unsuitable due to excessive soft tissue bulk, prompting the surgeon to opt for an alternative strategy during the surgical procedure.

Overall, among the 47 procedures analyzed, we noted 11 interventions with complications, 5 of which were major, including two surgical site infections necessitating further surgical debridement (4.3%; 95% CI: 0.5–14.5%). When compared to a group of 58 children (63 total procedures) where surgery was conducted solely with VSP, we observed 15 complications, 8 of which were major (with no infections). Based on the available data, the intraoperative use of 3DP-PSI did not significantly raise the infection rate or cause other complications (*p*-value > 0.05).

## 4. Discussion

To our knowledge, this is the largest study examining surgical outcomes using in vivo ann-HTPLA 3DP-PSIs in children. Among the 83 cutting guides produced, 79 effectively supported planned surgical interventions without significant adverse reactions or unexpectedly high complication rates associated with their use.

A previous study conducted by our group also showed a decrease in the average duration of surgeries utilizing 3DP-PSIs by about 45 min. This decrease is accompanied by a decline in postoperative complications and a reduction in the use of intraoperative fluoroscopy [19].

We observed that most of the jigs have consistently met expectations, yielding excellent clinical outcomes characterized by reduced operating time and decreased reliance on intraoperative fluoroscopy. Surgeons also report enhanced confidence in execution. The design and printing parameters, albeit determined empirically, have consistently delivered satisfactory results thus far. Nonetheless, it is crucial to highlight areas for optimization moving forward.

The primary concern is assessing acceptable geometric deviations induced by warpage between the designed object and final product post-3D printing, annealing, and sterilization while ensuring proper surgical device usability.

Starting from the creation of the virtual 3D bone model all the way to the fabrication of the cutting guides, it is important to emphasize that everything is customized to the anatomy of each individual patient. In fact, a crucial element is the presence of a CT scan of the patient, to begin with.

As for the design process and the assessment of the surgical parameters such as K-wire and slots for saws positioning and parameters before printing, Popescu et al. defined some guidance for the dimensioning parameter of these features, with which we agree. However, they report that it is advisable to consider a minimum span of 160° of the guide over the bone, especially for long bone with a predominant cylindrical shape, to ensure a good fit [28]. From our experience, as we validate their assertions, it is pertinent to highlight that dealing with small bone segments, especially in pediatric surgical procedures, poses unique challenges compared to the most common adult cases. To ensure precision, we aim to keep cutting templates as compact as possible and to utilize 3D printing for the relevant bone surface, complete with pre-marked holes and osteotomy lines. This allows for a verification of the proper adherence of the guide to the bone and checking the passage of K-wires via the holes and of saw blades via the slots to ensure that no printing errors or deformations due to heat treatment have occurred affecting the use of the guide. In summary, after design and printing, the guides are meticulously cleaned from printing supports, empirically tested, heat-treated, and repositioned on the model for final verification before sterilization.

We observed only one case of significant warpage and geometric deviations from the initial design affecting the proper use of the 3D-printed devices on the patient’s bone. We are uncertain if this was due to printing settings or the surgery being performed months after PSI production, which underwent double sterilization due to intervention postponement. However, we acknowledge the presence of an average geometric deviation between the design and the sterilized product. Frizziero et al. found an average deviation of 1.81% in the ann-HTPLA 3D-printed tibial cutting guide from the initial design, with a maximum absolute deviation of 4.85%, equivalent to 0.97 mm [23]. Moreau et al. reported in a recent study that HT-PLA exhibited superior performance compared to other types of PLA, with minimal deformation of 0.6 mm on 30 mm edge cubes (2%) and only 0.02 mm overall deformity on cuboid bone models measuring approximately 36 mm, even after standard steam sterilization [15]. Both studies confirm an average deviation of less than 1 mm from the initial design to the final ann-HTPLA 3D-printed bony models or PSIs, after undergoing steam sterilization. Based on our experience, we find that this average geometric deviation of 2%, typically less than 1 mm compared to the original dimensions, does not significantly impact mask fit, mask-bone alignment, or the use of metallic objects like K-wires or oscillating saws, commonly used in pediatric orthopedic surgery.

In our study, due to the diverse cases and numerous cutting guide designs utilized, we lack data on the average geometric deviation of each device post-annealing and sterilization. Consequently, we cannot definitively ascertain whether an average percentage of deviation of 2% or less, or less than 1 mm, is clinically significant enough to deem the deviation acceptable.

The degree of warpage and accuracy of parts produced by FDM significantly depends on both the size and shape of the designed object and the process parameters utilized [12].

In particular, the size of the 3D-printed product is a critical aspect. In our experience with children, we have never printed guides longer than 198 mm due to a 270 × 200 × 200 mm printing limit. This constraint may affect using larger custom devices. Yet, considering the average volumes of our printed objects, approximately 12 ± 8 cm^3^, we found no warpage significant enough to compromise the device’s usability or the execution of the planned surgery. Moreover, identifying the shape and dimensions of specific object parts, like holes and slots, is another critical aspect. Using testing, we established reference values suitable for commonly used surgical instruments, yet validated parametric methods to optimize diameter tolerances for wires and blade sizes were lacking.

Concerning the printing parameters, recent research emphasizes the critical role of the object’s printing position on the build platform, with most deformation occurring along the *X*-axis due to the intrinsic anisotropy of 3D-printed objects resulting from the FDM method [14]. Printing direction is also key; for instance, ann-HTPLA demonstrates the highest tensile strength when printed with a raster angle of 45/−45° [23]. Infill is another important factor. We have consistently used 100% infill, ensuring maximum strength and minimal heat deformation. In Grubbs et al.’s study, it was found that the infill density and pattern had a substantial effect on the mechanical properties of 3D-printed parts, resulting in a notable decrease in both strength and ductility when the infill density fell below 100% [29]. Yet, future research may reveal that optimal printing quality can be achieved with lower infill, speeding up the process and reducing material usage [18].

Another concern arises from the slight increase in the surgical site infection rate observed in patients treated with 3DP-PSI compared to those treated with VSP alone. Although not statistically significant and likely influenced by other patient factors and comorbidities, it is possible that 3DP-PSIs contributed to the infection rise due to imperfect sterilization with standard cycles. Our study continues, as a power analysis suggested analyzing at least 366 surgeries (183 with and 183 without using 3DP-PSIs) to ascertain the statistical significance of this difference in proportions, with a 5% alpha error and 80% statistical power. In their study, Ferràs-Tarragó et al. demonstrated that 3D-printed PLA objects can be effectively sterilized using steam at 134 °C, achieving 100% sterilization effectiveness without any bacterial growth post-sterilization [18].

Further investigation will be conducted on the heat treatment phase. As mentioned earlier, previous studies had already been conducted to select the best material, which turned out to be ann-HTPLA [15,23].

Furthermore, although cutting guides are not designed to endure specific forces or loads, their main function is to indicate cutting angles and positions for replicating the surgical plan in vivo based on the virtual patient-specific model. Nonetheless, it is noteworthy that the annealing process, when executed under certain parameters, can also improve other properties, such as tensile strength. Indeed, Suder et al. indicate a maximum tensile strength obtained, on average, at 62.94 MPa with a material annealing temperature of 80 °C, which was, on average, 9.62% higher compared to non-annealed samples [30].

To make a leap in quality, it will be necessary to anticipate warpage and other minor mechanical changes that may arise throughout the entire 3D printing, annealing, and sterilization process. This approach reduces the risk of 3DP-PSI deformation during treatment and pre-emptively addresses potential issues by adjusting object design and printing parameters. Currently, preliminary experiences have utilized finite element models to forecast the thermomechanical behavior of 3D-printed objects during printing [13].

The aforementioned applies to 3DP-PSIs crafted from ann-HTPLA using FDM technology. It is evident that alternative materials and printing technologies may overcome or complement the limitations of our current approach.

### Limitations

Our study has limitations. Firstly, while we found no issues with utilizing the 3DP-PSIs in most cases, we did not measure sterilized devices during surgery for warpage and geometric deviations. Previous in vitro studies by our group showed minimal deviations in printed and sterilized objects compared to the designed ones [23]. Collecting such data during surgery is challenging, requiring sterile calipers and minimizing guide manipulation to avoid prolonging surgery.

Another limitation of this study is the lack of quantitative parameters to evaluate the usability of 3DP-PSIs. Although we examined significant obstacles that have impeded the proper use of the guides, it remains necessary to establish a system or define quality metrics to ensure the quality and usability of the final product.

Finally, as we have consistently employed cutting guides solely for replicating surgical plans, certain factors like roughness and force resistance have been overlooked. To advance and broaden the application of patient-specific instruments, we plan to explore various associated properties in forthcoming studies.

## 5. Conclusions

In conclusion, ann-HTPLA has emerged as a dependable and safe material for 3DP-PSIs in pediatric orthopedic surgery. Utilizing FDM technology for 3D printing guides in HTPLA, along with post-processing thermal treatment, offers surgeons reliable custom tools while enabling convenient, cost-effective use in an in-office setting with minimal setup requirements, ensuring safety for both the final product and the operator. Nevertheless, further studies are necessary to enhance and standardize the production process.

## Figures and Tables

**Figure 1 polymers-16-01216-f001:**
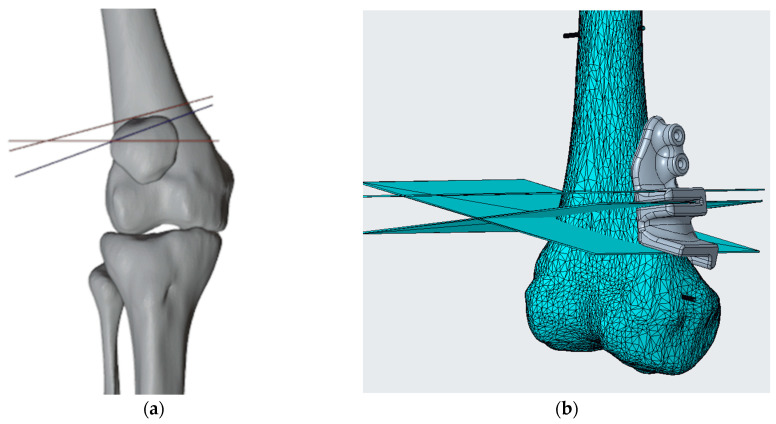
(**a**) Virtual surgical planning phase with identification of the location of cutting planes that may also have different purposes, depending on the planning needed (in the example in the figure you can see the plans in red for making the osteotomy plans to realize the correction, in blue, the plane for shaping the bone graft for use as an autologous graft) and K-wires for guide fixation in Blender. (**b**) Design of the cutting guide in Creo Parametric.

**Figure 2 polymers-16-01216-f002:**
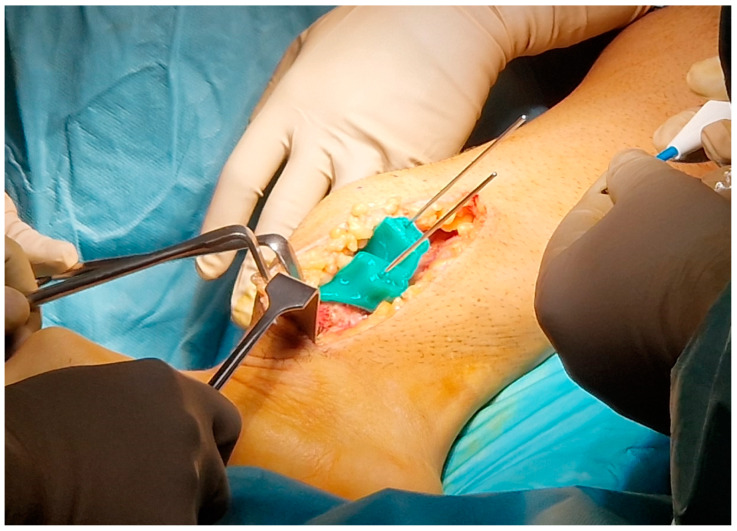
Clamping by K-wires of the cutting mask during surgery.

**Figure 3 polymers-16-01216-f003:**
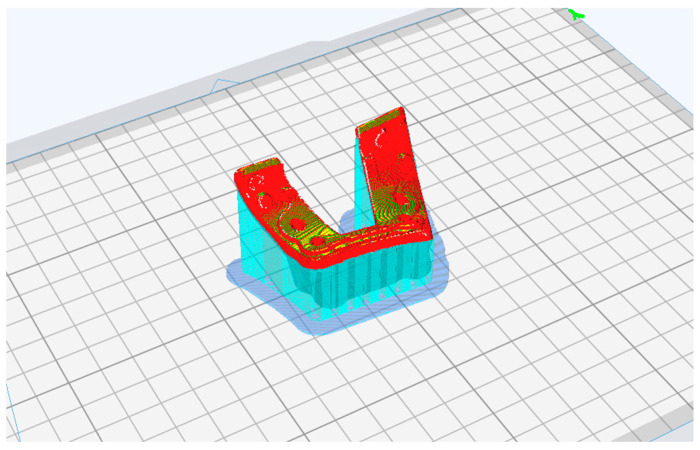
In preparing the 3D-print of the cutting guide, ensure that the side intended to make contact with the bone remains curled upward. This prevents unnecessary support material from being applied to the critical bone-contacting surface.

**Figure 4 polymers-16-01216-f004:**
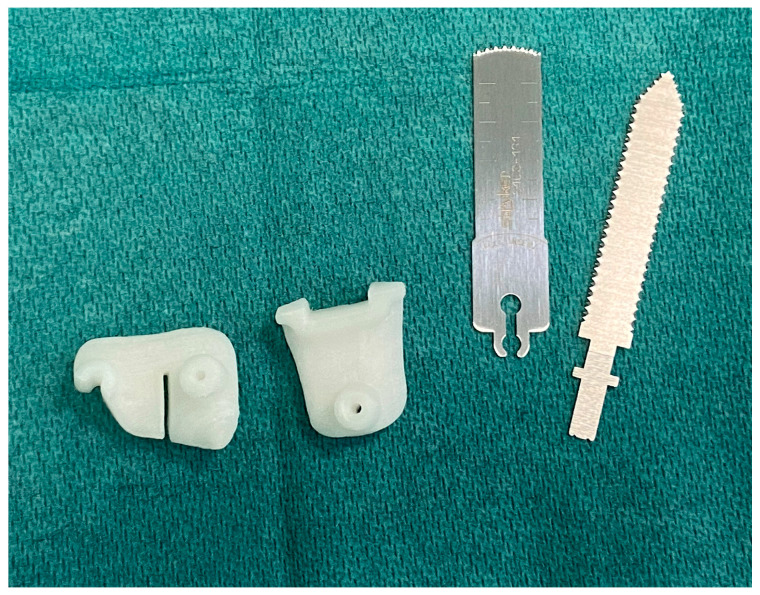
Sterile cutting guides are brought directly to the operating room, ready to be used as support for standard instrumentation.

**Figure 5 polymers-16-01216-f005:**
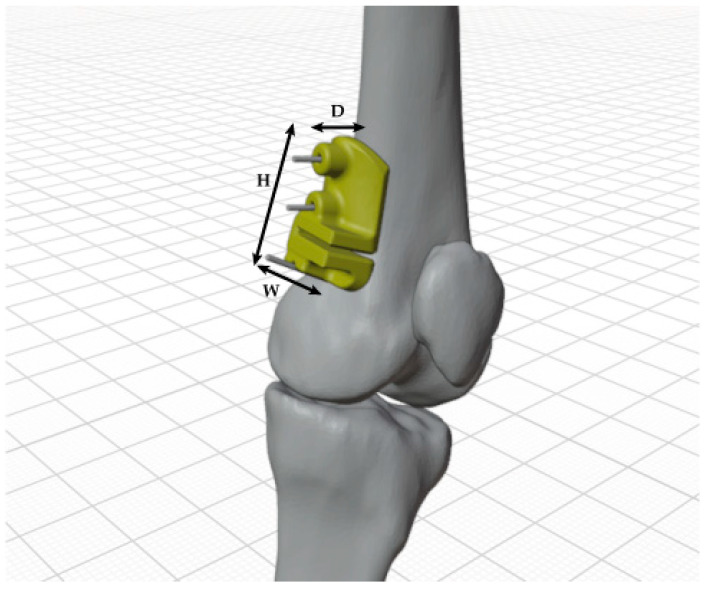
Example of the dimensioning (H: height; W: width; and D: depth) made for cutting guides.

**Figure 6 polymers-16-01216-f006:**
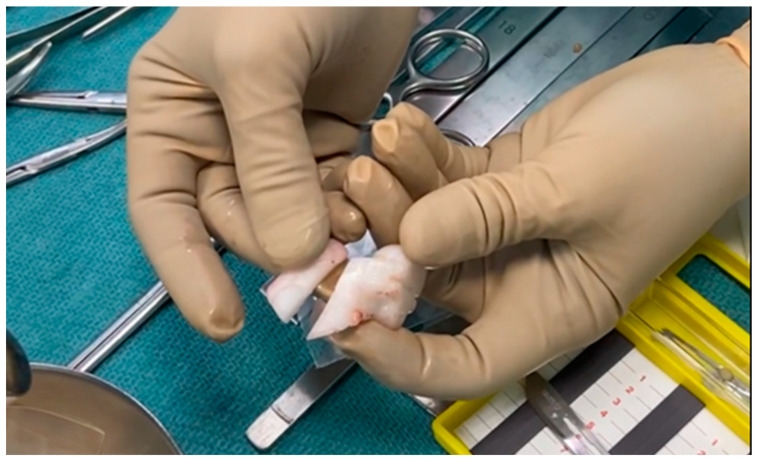
Broken cutting guide along the direction of 3D printing layers.

**Table 1 polymers-16-01216-t001:** Main PLA Crystal Clear (Fillamentum Manufacturing Czech s.r.o.) properties.

PLA Crystal Clear		
Thermal properties	Glass transition temperature	55–60 °C
	Melting point	150–230 °C
	Decomposition temperature	>230 °C
Printing properties	Print temperature	210–230 °C
	Hot pad	50–60 °C
Physical properties	Material density	1.24 g/cm^3^
Mechanical properties	Tensile strength	50 MPa

**Table 2 polymers-16-01216-t002:** Printing parameters used for the fabrication of the cutting guides.

Parameter	Value
Nozzle width	0.4 mm
Layer height	0.2 mm
Infill	100%
Infill overlap percentage	20%
Nozzle temperature	220 °C
Bed temperature	60 °C
Speed	60 mm/s
First layer speed	15 mm/s

**Table 3 polymers-16-01216-t003:** Times and temperatures of the thermal cycle used to annihilate HTPLA.

Material	Cycle	Time	Temperature
HTPLA	1st cycle	10 min	80 °C
	2nd cycle	50 min	100 °C

**Table 4 polymers-16-01216-t004:** Geometric dimensions and printing parameters of 3D-printed cutting guides. Values are expressed in raw numbers, means ± SD, and range.

Anatomical Site	N.	Height (mm)	Length (mm)	Depth (mm)	Volume (cm^3^)	Filament Length (m)	Filament Weight (g)	Printing Time (min)	Layers (Number)
Upper limb	4	31 ± 11	34 ± 17	17 ± 7	5 ± 3	3 ± 1	8 ± 4	53 ± 20	90 ± 33
		(23–48)	(21–58)	(7–25)	(4–10)	(2–5)	(6–14)	(39–82)	(47–123)
Lower limb	79	54 ± 30	36 ± 11	32 ± 26	12 ± 8	7 ± 4	20 ± 12	122 ± 61	147 ± 58
		(20–198)	(11–63)	(3–156)	(1–59)	(1–25)	(2–74)	(14–317)	(19–304)
Total	83	53 ± 30	36 ± 11	31 ± 26	12 ± 8	7 ± 4	19 ± 12	118 ± 61	144 ± 58

## Data Availability

The data presented in this study are available on request from the corresponding author. The data are not publicly available due to privacy and ethical reasons.

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
