# Peer review of "High-Temperature Polylactic Acid Proves Reliable and Safe for Manufacturing 3D-Printed Patient-Specific Instruments in Pediatric Orthopedics—Results from over 80 Personalized Devices Employed in 47 Surgeries"

_polymers, 2024, doi:10.3390/polym16091216_

Round 1
Reviewer 1 Report
Comments and Suggestions for Authors
This article covers the important topic of using the 3D printed parts in surgery. The overall article experimental design is good, introduction provides the complete background of the topic, experiments are well-described and the scientific design is very good, conclusions are supported by results and the discussion covers most of the problems. I would like to suggest to extend a bit discussion towards the requirements for printed parts and printing including roughness and possible defects and trace it into the requirements to printer device, what pre-checks should be done and similar things (some checklist before using the printed part).
Comments on the Quality of English LanguageEnglish language is good except some non-parallel sentences like "This includes
reduced operating time, fluoroscopy, bleeding, and complications"
Author Response
Dear Reviewer,
Thank you very much for your time and suggestions.
Until now, cutting guides have only been employed as pure guides for replicating surgical planning performed in CAD virtual environment, neglecting parameters like roughness and force resistance, as they were not intended to bear loads. To enhance and broaden the utilization of patient-specific tools, we plan to explore associated properties in upcoming studies. Thank you very much, Your suggestion served as inspiration for us to incorporate it into the study limitations.
As suggested, we have expanded the discussion by providing an explanation of how we empirically verify each cutting guide and some guidelines found in the literature. We also added a supplementary material for the checklist.
Also, we have modified the sentence you pointed out "This includes reduced operating time, fluoroscopy, bleeding, and complications" by correctly integrating it into the speech.
Best regards,
The authors
Reviewer 2 Report
Comments and Suggestions for Authors
3D-printed, customized equipment have revolutionized orthopedic surgery by improving accuracy and lowering complications. High-Temperature PLA (ann-HTPLA) can be annealed to exceed the temperature restrictions of PLA, allowing for steam sterilization without sacrificing characteristics. The in-vivo effectiveness of ann-HTPLA 3DP-PSI in juvenile orthopedic surgery is investigated in this work. The authors examined the safety and effectiveness of utilizing ann-HTPLA 3DP-PSI to treat pediatric patients' abnormalities of the limbs. Dimensions of the 3DP-PSI, printing parameters, usage, and complications were all recorded.
Paper is nice structured and well-done written. Figures are appropriate. IMRAD stucture is adopted. And the paper corresponds to Aim and Scope of the Journal. It also should be mention that it has a very promising technique for medical applications.
Nevertheless, there are small comments:
1. As I correctly understood, you do not use patient-specific models, but you use customized model made in Blender software. If you use MRI images of patients and process them in a special software, please mention it. So, if you use a customized model, it can be a kind of limitation. Please highlight it in Materials and Methods. And 2-3 sentences in Discussion.
2. It would be nice to discuss more elaboratly printing regimes. And more precisely consider it in Materials and Methods. Nozzle? Printing rate? Another features. How it affect the strentgh after high-temperature sterilization?
3. Not much refernces in Discussion section.
So, in my opinion a paper can be published with minor corrections.
Author Response
Dear Reviewer,
Thank you very much for your time and suggestions.
We made some corrections as you suggested.
- We use patient-specific CT scans to generate, through a process known as segmentation, the 3D patient-specific model for both surgical planning and cutting guides fabrication. In the materials and methods and discussion sections, we specified that the process starts with CT scan processing as you suggested.
The segmentation process is a widely recognized procedure in the literature and previously detailed in our own studies and is referenced accordingly. Please indicate if you require further elaboration. - We have added a few more printing parameters in the materials and methods section. Furthermore, we included a comment regarding the effect of annealing on tensile strength in the discussion for completeness improving also references in Discussion section as asked in point 3.
- Improved as suggested.
Best regards,
The Authors